# Geopolymer Concrete: A Material for Sustainable Development in Indian Construction Industries

**Manvendra Verma** [1], **Nirendra Dev** [1], **Ibadur Rahman** [2], **Mayank Nigam** [3], **Mohd. Ahmed** [4,*] and **Javed Mallick** [4]

1    Department of Civil Engineering, Delhi Technological University, Delhi 110042, India; manvendraverma_2k16phdce05@dtu.ac.in (M.V.); nirendradev@dce.ac.in (N.D.)
2    Department of Civil Engineering, Jamia Millia Islamia, Delhi 110020, India; irahman1@jmi.ac.in
3    Department of Civil Engineering, GLA University, Mathura 281406, India; mayank.nigam@gla.ac.in
4    Civil Engineering Department, College of Engineering, King Khalid University, Abha 61421, Saudi Arabia; jmallick@kku.edu.sa
*    Correspondence: mall@kku.edu.sa

**Abstract:** Geopolymer concrete (GPC) is a new material in the construction industry, with different chemical compositions and reactions involved in a binding material. The pozzolanic materials (industrial waste like fly ash, ground granulated blast furnace slag (GGBFS), and rice husk ash), which contain high silica and alumina, work as binding materials in the mix. Geopolymer concrete is economical, low energy consumption, thermally stable, easily workable, eco-friendly, cementless, and durable. GPC reduces carbon footprints by using industrial solid waste like slag, fly ash, and rice husk ash. Around one tonne of carbon dioxide emissions produced one tonne of cement that directly polluted the environment and increased the world's temperature by increasing greenhouse gas production. For sustainable construction, GPC reduces the use of cement and finds the alternative of cement for the material's binding property. So, the geopolymer concrete is an alternative to Portland cement concrete and it is a potential material having large commercial value and for sustainable development in Indian construction industries. The comprehensive survey of the literature shows that geopolymer concrete is a perfect alternative to Portland cement concrete because it has better physical, mechanical, and durable properties. Geopolymer concrete is highly resistant to acid, sulphate, and salt attack. Geopolymer concrete plays a vital role in the construction industry through its use in bridge construction, high-rise buildings, highways, tunnels, dams, and hydraulic structures, because of its high performance. It can be concluded from the review that sustainable development is achieved by employing geopolymers in Indian construction industries, because it results in lower $CO_2$ emissions, optimum utilization of natural resources, utilization of waste materials, is more cost-effective in long life infrastructure construction, and, socially, in financial benefits and employment generation.

**Keywords:** geopolymer concrete; fly ash; GGBFS; compressive strength; mechanical properties; durability properties and bond strength

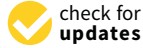



## 1. Introduction

After water, concrete is the world's second most abundant resource. Portland cement was traditionally used as a binding material in concrete. In 2018, India manufactured approximately 502 million tonnes of cement. Portland cement is not environmentally sustainable, and there are many environmental concerns involved with its manufacturing. Cement processing accounts for about 8% of greenhouse gas emissions from carbon dioxide [1]. Cement development depletes renewable resources and pollutes the climate, endangering our future [2]. We must pursue sustainable development because the earth's natural resources are limited [3].

On the other hand, there is an excess of agricultural solid waste on the planet, such as fly ash, slag, and rice husk ash. In India, thermal power plants generate 200 million tonnes

of fly ash per year, while steel manufacturing plants produce 20 million tonnes of slag per year. Fly ash and slag are wastes from that sector; they deposit industrial solid wastes on cultivated land and contaminate the atmosphere due to the restricted mode of industrial solid wastes [4].

The geopolymer was introduced by the Davidovits in 1978 and is made by the alkaline solution's activation (consists of sodium or potassium silicate and sodium or potassium hydroxide) of the high alumina silica-rich materials. It is like ceramic composites that make the bond between alumina and silica. Geopolymer-based concrete using fly ash has a high potential for the construction industry to replace ordinary Portland Cement (OPC)-based concrete with comparable structural properties [5]. Figure 1a describes the fly ash generation and utilisation from 1996 to 2019 and data from the mineral book 2019. Figure 1b shows the description of the fly ash used in various departments in 2019 with their percentage utilisation [6].

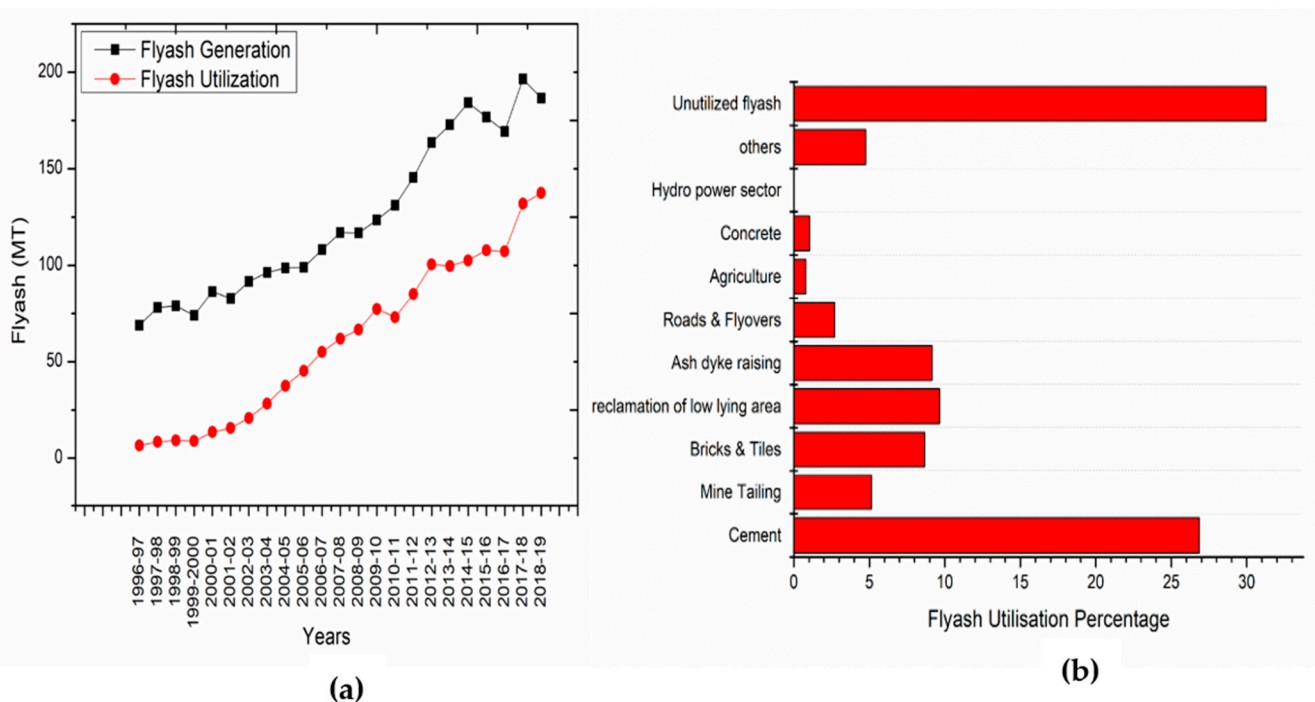

**Figure 1.** (**a**) Fly ash generation and utilisation in India 1996–2019; (**b**) Utilisation of the fly ash in various in India 2019 [6].

Geopolymerisation is a reaction in which the formation of the alumino-silicate gel structure uses silica and alumina content from the pozzolan binding materials like fly ash and slag activated by an alkaline solution (NaOH and $Na_2SiO_3$). Figure 2a shows the mechanism of geopolymerisation, describing the constituents and their effect on the end products. The microstructure of the crosslinked geopolymer shows higher stability against external environmental conditions. In the geopolymerisation process, alkali metal plays an essential role in the reaction rate and end products. It is an inorganic polymer that shows an amorphous nature at high temperatures. Heat-curing increases the geopolymerisation rate compared to ambient-temperature curing. Equations (1) and (2) show the geopolymerisation reaction process from starting to the specimen hardening [7]. The end products of the geopolymerisation show a three-dimensional structure.

$$n\,(Si_2O_5,\ Al_2O_2) + 2nSiO_2 + 4nH_2O \ \rightarrow \ n\,(OH)_3\text{–}Si\text{–}O\text{–}Al^{(-)}\text{–}O\text{–}Si\text{–}(OH)_3 \atop \qquad\qquad\qquad\qquad\qquad\qquad |\ \ (OH)_2 \tag{1}$$

$$n\,(OH)_3\text{–Si–O–Al}^{(-)}\text{–O–Si–}(OH)_3 \;\rightarrow\; (Na^+, K^+)\text{–}(\text{–Si–O–Al}^{(-)}\text{–O–Si–O–}) + 4nH_2O \qquad (2)$$

The alkali metals like Na and K are primarily used in the activator solution to activate the pozzolanic binding material, and they form the zeolitic structure during the geopolymerisation reaction. The $SiO_2/Al_2O_3$ ratio plays a vital role in forming the microstructure at the curing temperature. Figure 2b shows the curing temperature's effects on the $SiO_2/Al_2O_3$ ratio forming the end product's zeolitic structure. It also shows the gel product's amorphous nature at elevated temperatures by forming a poly-sialate-siloxo $(Si\text{-}O\text{-}Al\text{-}O\text{-}Si\text{-}O\text{-})_n$. The content present in the mix design does not involve the end product of the geopolymerisation reaction.

### 1.1. Research Significance

Geopolymer concrete is economical, low energy consumption, thermally stable, easily workable, eco-friendly, cementless, and durable. GPC reduces carbon footprints by using industrial solid waste like slag, fly ash, and rice husk ash. Around one tonne of carbon dioxide emissions are needed to produce one tonne of cement; these emissions directly pollutes the environment and increases the world's temperature by increasing greenhouse gas production. For sustainable construction, GPC reduces the use of cement and presents an alternative to cement for the material's binding property. GPC has better physical, mechanical, and durable properties than Portland cement concrete and it is highly resistant to acid, sulphate, and salt attack. Thus, geopolymer concrete is an alternative to Portland cement concrete and is a material with a potentially large commercial value for sustainable development in the construction industry. The study presents the state-of-art review of the GPC production, its performance (environmental, socio-economic) and applications as a material that can be useful for sustainable development in construction industry. This review is an effort to join and share the scientific and technical information, and to bridge the knowledge gap in GPC utilization as a material for sustainable development in construction industry.

### 1.2. Definition of Abbreviations Used in the Manuscript Script

Geopolymer concrete (GPC); plain cement concrete (PCC); ordinary portland cement (OPC); ground granulated blast furnace slag (GGBFS); air-cooled slag (ACS); Alkali-Activated Fly ash/Slag (AFS); Alkali–Silica Reaction (ASR); Biomass Fly ash (BFA); Bottom Ash (BA); alkali-activated fly ash (AAFA); fibre-reinforced plastic (FRP); GFPR (glass fibre reinforced polymer); fly ash–slag geopolymer concrete (FSGC); high-range water-reducing admixture (HRWRA); high volume fly ash (HVFA); high volume slag (HVS); light weight aggregate (LWA); lightweight expanded clay aggregate (LECAs); microencapsulated phase change materials (MPCM); recycled concrete aggregates (RCA); self-compacting geopolymer concrete (SCGC); naphthalene formaldehyde (SNF); polycarboxylate ether (PCE); super plastically formed (SPF); high performance ash (HFA); rice husk ash (RHA); C-S-H (Calcium Silicate Hydrate (C-S-H); calcium alumino-silicate hydrate (C-A-S-H); calcium-sodium aluminosilicate hydrate [C-N-A-S-H]; energy dispersive X-ray analysis (EDXA).

### 1.3. Research Methodology

The main aim of this research is to review GPC as a material for sustainable development in the construction industry of developing countries, i.e., material that satisfies environmental (high performance, durability, and eco-friendly applications) and socio-economic (society acceptance, commercial, and cost) aspects of sustainability. The research methodology is based on the processing of the secondary data, i.e., published research and books. The secondary data are taken from the databases of scientific articles, including Scopus, ScienceDirect, and Google Scholar. It is focused on the keyword "geopolymer

concrete", and is supported by other key words; for example, "mechanical properties" and "durability properties".

## 2. GPC Manufacturing Studies

Geopolymer mortar and concrete require water and superplasticiser to increase workability because the sodium silicate and NaOH concentration increment degrades the fresh mortar's and concrete's flowability and workability; usage of superplasticizer also harms the strength of the geopolymer mortar [8]. The slag and sodium silicate dosage increment in the GPC mix reduces the porosity and simultaneously reduces the mesopore volume, and creates a denser matrix, due to the more reactive products. The autogenous shrinkage of geopolymer mortar develops due to the self-desiccation in the solid-state but chemical shrinkage in the fresh state. The AFS mortar shows higher drying shrinkage than the OPC specimens due to the mixed samples' high capillary stress [9]. The AFS mortar has higher compressive strength, flexural strength, and lower water absorption than the OPC mortar samples. The hydration product of the AFS paste is mostly amorphous, and the higher fly ash content shows less length change [10].

The water to binder ratio increment in the AFS mortar increases the flow and workability of the mortar. A fine aggregate to binder ratio above 2.5 results in the instantaneous degradation in the flow of mortar. The GGBFS-based mortar shows a higher strength than the fly ash-based mortar, and the compressive strength of the mortar increases with the increase of the fine aggregate to binder ratio up to 2.5. Beyond 2.5, the strength drastically degrades [11]. The geopolymer mortar strength increases with the rise in temperature, but it declines beyond the 800 °C temperature. The geopolymer mortar compressive strength depends on the bonding between the binder and aggregates, and the increment aggregates in the mortar reduce the geopolymerization reaction [12]. The thickness of the geopolymer paste is the main factor influencing the mortar's heat-resistant property, and changes in aggregate and the mass ratio of paste to fine aggregate affect the paste's thickness. If it is too thick, the water vapour can collapse the structure when released, and if it is too thin, the paste cannot bond aggregate together.

High calcium fly ash usage in the mix provides a high-strength geopolymer mortar up to 86 MPa at 28 days [12]. The heat evolution during the geopolymerization is first due to sodium silicate and gelation, and second due to bulk hydration of GGBFS, responsible for the strength development observed in the calorimeter. After the reaction, the products are amorphous and uniformly distributed throughout space in the matrix [13].

### 2.1. Fly Ash Based GPC

Fly ash rich in silica and alumina constituents shows pozzolanic characteristics and binding properties. The fly ash is used in the GPC, which provides better workability and higher solid to liquid ratios than the other pozzolanic materials like slag, metakaolin, and rice husk ash due to its small particle size, and spherical and porous properties [14]. In the GPC, the bottom ash usage increases the strength and workability of the increasing surface for reactivity and collapses the pores in the matrix due to their higher fineness [15]. The increase in ultrafine fly ash dosage decreases the setting time, but up to 15% dosage, ultrafine fly ash usage and the setting time are longer for the fly ash-based GPC. The use of ultrafine fly ash increases by 10%, reducing the matrix's porosity of the GPC [16]. The fineness of fly ash is the deciding factor for the amount of fly ash required for the mix design of GPC, because increasing the fineness of fly ash decreases the quantity requirement for mix design [17]. The fineness of fly ash plays a vital role by reducing the porosity and water absorption capacity of both concrete GPC and OPC concrete [18]. The fineness of the fly ash increases the workability, density, and strength of the GPC. It slightly affects the alkalinity of the GPC but is similar to the OPC concrete [19]. Figure 3a shows the effect of the fineness of the fly ash on the density of the concrete mix. The GGBF/fly ash ratio is 20/80, which is the optimum point of mechanical strength [20].

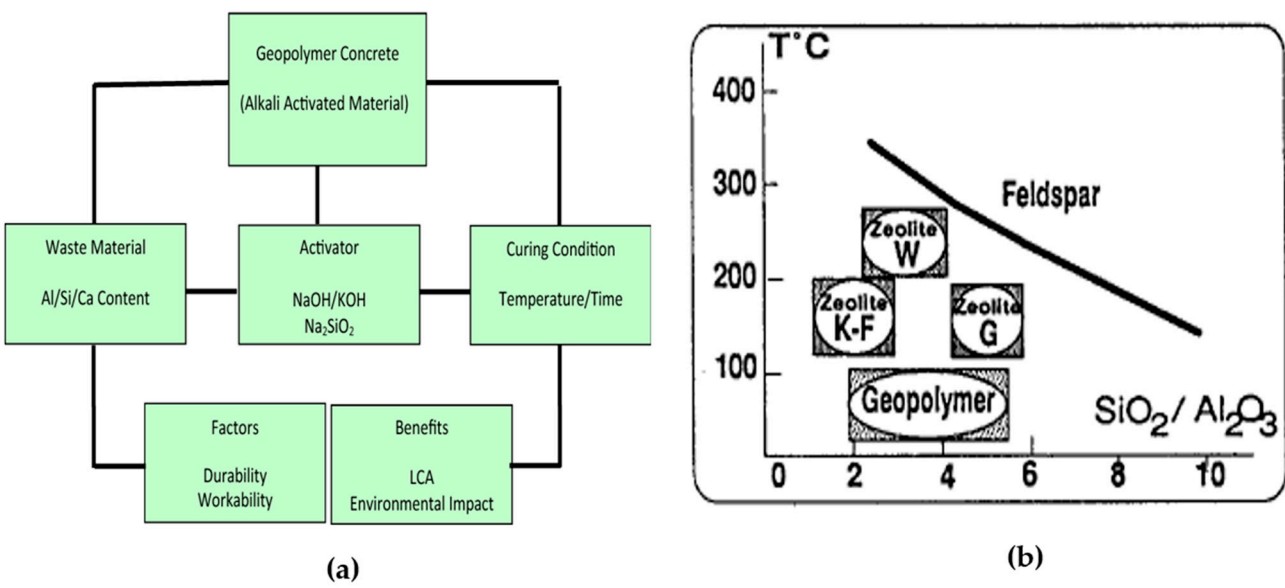

**Figure 2.** (**a**) Mechanism of Geopolymerisation; (**b**) crystallisation temperature ranges for $K_2O$-$Al_2O_3$-$SiO_2$ system [21].

### 2.2. GGBFS Based GPC

In the GPC, increasing the dosage of GGBFS decreases the mix's workability due to the angular particles of samples compared to the spherical particles of fly ash. It decreases the ambient temperature curing setting time due to the reaction of calcium present in the slag. The slag's usage with fly ash in the GPC presents a similar compressive strength to ambient-cured specimens to the water-cured OPC concrete. The specimen strength gains slow down after 28 days but strengthen up to 180 days [22]. In the GPC, the strength increases with the dosage of GGBFS in the mix design and eliminates heating conditions for curing [23]. The mix design's slag/fly ash ratio increases, showing less mass loss at the elevated temperature [24]. The heat-cured fly ash-based GPC gives a higher strength, but this could be achieved by replacing some fly ash with the GGBFS at ambient-curing conditions [25].

### 2.3. Effect of Molar Ratios of Alkaline Solution

The compressive strength of GPC increases with an increase in the molarity of sodium hydroxide, but the workability declines [26–31]. The GPC uses sodium hydroxide to increase the mechanical properties compared to the OPC concrete, which results in its higher strength and performance [26,32]. Instead of the sodium silicate, KOH generates a much more heterogeneous structure, with higher porosity and lower strength developed after hydration [13]. Sodium hydroxide with sodium silicate is used as an alkaline activator of the binder in a 1:1 ratio in the mix design to produce GPC strength of 40 MPa and above, and verify the potential of fly ash to replace the cement [33]. The fly ash/slag-based paste setting time reduces as the slag's dosage, sodium silicate, and sodium hydroxide molarity increase. The initial time is 55 min, and the final time is 160 min for the fly ash/slag-based paste at a room temperature of 17 °C [34]. The compressive strength of 30 MPa was quickly achieved with the range of molarity of 9.5–14 M of NaOH at room temperature in 28 days [35]. The geopolymer matrix developed in GGBFS is similar to the matrix developed in the absence of GGBFS. If the NaOH concentration is low, the calcium content of the GGBFs participates in the development of amorphous CSH gel and the formation of calcium-based geopolymer. If the concentration of NaOH is high, the precipitation of calcium hydroxide develops [36].

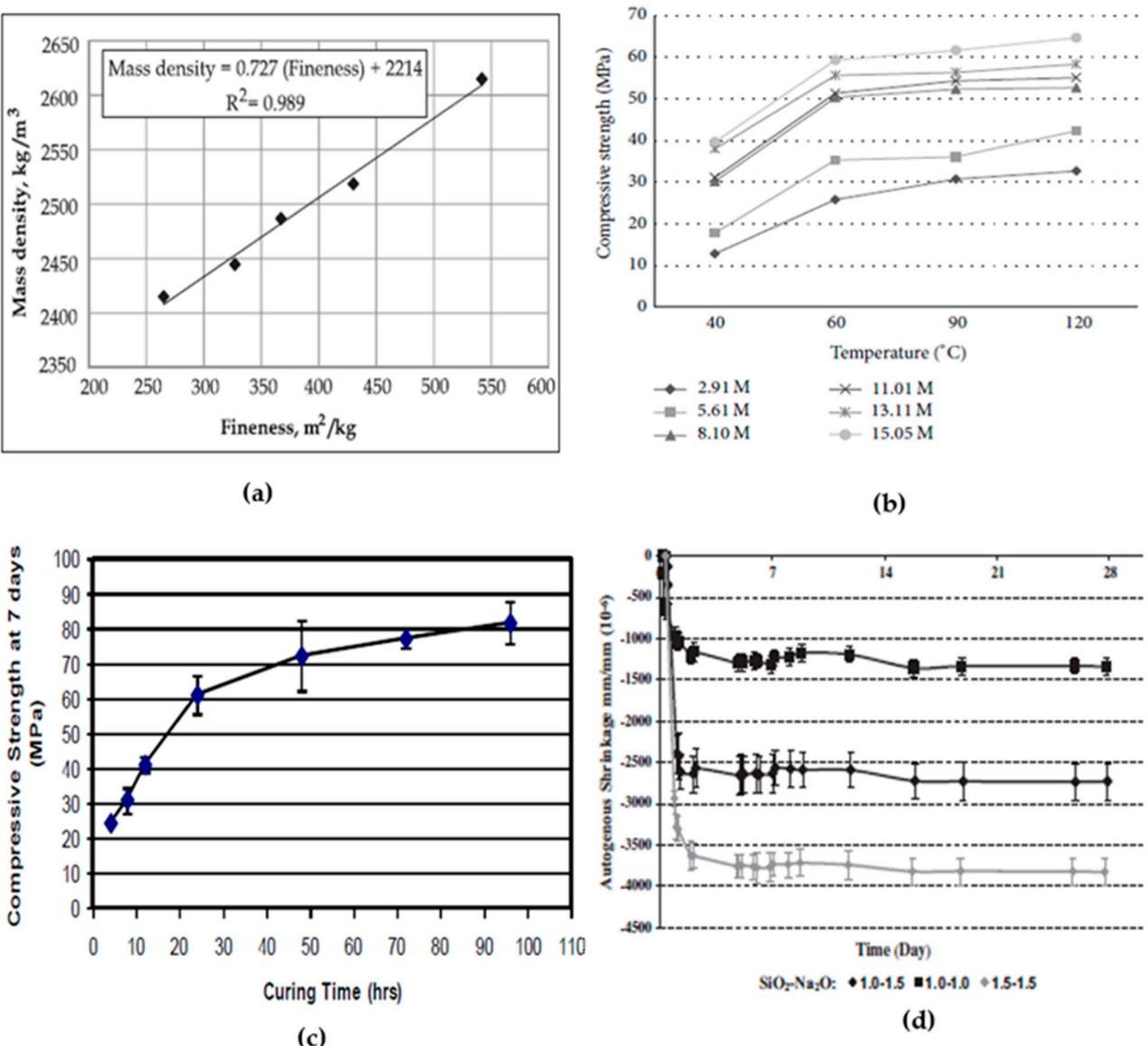

**Figure 3.** (**a**) Effect of fly ash fineness on mass density; (**b**) effect of degree of heating for different concentrations of NaOH solution on compressive strength at solution-to-fly ash ratio of 0.40 [27]; (**c**) effect of curing time on compressive strength [30]; (**d**) autogenous shrinkage of AAFA pastes with different SiO$_2$ and Na$_2$O content, cured at 40 °C (corrugated tube method, measurement started after the final set, the autogenous shrinkage was the mean value of three replicates for each specimen).

The addition of sodium silicate extensively increases the hybrid geopolymer system's early strength development [37]. The interface between aggregate and geopolymeric paste is not apparent after the high soluble silicate dosage in the salt-free geopolymeric mortar. In the geopolymerisation process, soluble silicates are very useful in degrading alkali saturation in the GPC pore solution even after using a high alkali concentrated activating solution and promoting the higher inter-particle bonding binder well as the aggregate surface. Geopolymerisation leads to reliable geopolymer results like binders, mortars, and concrete [38]. The ratio of sodium silicate to sodium hydroxide (by mass) increases with the increasing GPC compressive strength. If the alkaline solution content increases in the mix, then the setting time and workability increase with the reduction of the compressive strength. The alkaline ratio increased from 1.5 to 2.5, and the alkaline solution increased from 35% to 45% of the mass of binder, and the optimum point was found at 2.5 alkaline ratio and at 40% alkaline solution in the mix [39]. The Na$_2$CO$_3$ activator is

the cheapest activator in terms of the relative cost per ton and strength of geopolymer mortars [40]. The yield point of the compressive strength of mortar and paste found at the molar ratio of $Na_2O/SiO_2$ is 0.40 and shows the paste's denser morphology. If the $Na_2O/SiO_2$ molar ratio increases, then the consistency of the paste decreases. It forms an alumino-silicate gel with the fly ash content and is responsible for the higher workability and the mechanical properties of the GPC samples in the hardened state [41]. The increased ratio of $Na_2SiO_3/NaOH$ decreases the fine aggregate requirement in the mix design and increases the extra water requirement in the mix [17]. Figure 3b shows the relationship between compressive strength and curing temperature with various molarities of NaOH.

### 2.4. Effect of Calcination Temperature

The geopolymer manufactured from calcinated clays shows better mechanical properties than a geopolymer obtained from un-calcinated clays [42,43]. Calcination conditions (calcination temperature, heating rate, calcination time, atmosphere, and cooling rate), to achieve dehydroxylation of a given clay mineral, affect the pozzolanic reactivity, hydration kinetics, and consequently, strength and durability of geopolymer concrete [44,45]. The geopolymer derived from clay minerals calcinated at temperature of 900 °C shows better characteristics, such as compressive strength, water absorption, and stability in the aggressive environment of chloride ions, than the clay minerals calcinated at lower temperature than 900 °C [46].

### 2.5. Effect of Particle Size Fraction and Types of Aggregates

In the GPC, the aggregate size has a critical effect on the elevated temperature; the spalling of GPC reduces with the increase in the maximum aggregate size of the GPC mix design. The size of the fracture process zone increases with the aggregate size by shielding the crack tip. The maximum aggregate size increase in mix design provides higher resistance to fire by improving stability [47]. The GPC has better stability against elevated temperatures and better compatibility between the aggregates and pastes [48]. The geopolymer aggregate was developed from the mine tailings of fly ash and is used as a lightweight aggregate in mortars and concrete. The geopolymer aggregate shows better mechanical properties than the LECAs in the mix design solid state, whereas the rheological properties are the same. The geopolymer aggregate shows a better crushing value and excellent bonding due to its rough surface [49]. The geopolymer sand dosage in the mix design shows better mechanical properties up to 40% replacement of natural sand, but beyond 40%, the mix's mechanical properties decrease with the increasing content of geopolymer sand [50]. The concrete strength increases with the dosage of m-sand in the mix by replacing natural sand up to 20%, but beyond 20%, strength slightly decreases compared to the nominal mix [51]. The GPC uses 100% m-sand to replace natural sand, which shows adequate strength, and it is an economical, environment-friendly material that reduces carbon footprints by up to 80% [52]. In the GPC, the dosage of the RCA in the mix design by the partial replacement of the natural sand shows an adverse effect on the compressive strength, splitting strength, sorptivity, chloride ion penetration, water absorption, and volume of voids, but the properties are better than the OPC concrete [53]. The ductility of the GPC structural element is shown to increase with the partial replacement of natural sand in GPC design mix by waste aggregate of lime stone [54].

### 2.6. Effect of the Alkali Metal Activator

Geopolymer concrete is made by the geopolymerisation reaction among the binding constituents of the concrete, in which alkali metal cations play an important role in every stage of the reaction of geopolymerisation. The type of alkali metal cation is also an essential factor for the geopolymerisation reaction in all the reaction stages. It also affects the speed of the setting time of the paste and the condensation reaction of the GPC. The alkali metal cation works as a template for the reaction in directing and controlling the molecules' reaction and final structure formation. The size of the alkali metal cation also plays a role

in forming the structure of the geopolymer. The potassium cation (K) is responsible for a higher condensation than the sodium (Na) cation in the same condition. The potassium cation shows a higher compressive strength due to the higher surface area, produces a higher amorphous structure, and is less resistant to HCl attack. The alkali metal depends on the source material (Si and Al content) for the reaction because the different source materials directly affect the end products' physical and chemical properties [55].

The GGBFS-based geopolymer directly depends on the chemical composition of reactions. Increasing the sodium hydroxide (NH) and sodium silicate (NS) content in the composition increases the compressive strength, setting time and fire-resistant characteristics of the end products [56]. The geopolymer product's thermal stability with sodium-containing activators is less than activated potassium, due to functional changes in the microstructure. The sodium-activated geopolymer shows deterioration of strength sharply at 800 °C due to increases in the average pore size. The materials prepared with potassium silicate and fly ash show better thermal stability than the sodium-containing activator used. In the sodium activator, the amorphous structure is replaced by the crystalline Na-feldspar, but in the potassium activator used, the materials remain amorphous up to 1200 °C. The geopolymer material produced using fly ash and sodium or potassium silicate shows very high shrinkage and significant changes in the compressive strength on increasing the temperature range of 800–1200 °C [57].

In the geopolymer materials, a sodium-containing activator shows thermal stability up to 500 °C, which is more stable than the regular concrete mix [58]. The geopolymer's thermal shrinkage mostly depends on the Si/Al ratio and the alkali used in the activator. The low Si/Al ratio used in the mix shows better thermal stability than the higher Si/Al ratio and shows densification at elevated temperatures [59]. The alkali activator content also plays an effective role in finding the compressive strength of the GPC. Increasing the alkali content increases the compressive strength due to the solubility of the alumino-silicate with the increase in $Na_2O$ concentration, but beyond 10% it does not show a significant increase in strength. The content of $Na_2O$ ranges between 6–15% by mass. Sodium and silica content at a higher level show higher autogenous shrinkage than the cement matrix, but the autogenous shrinkage mechanism of the geopolymer matrix is different from the cement paste. In the geopolymer, the reorganisation and geopolymerisation of the structure develop a finer pore size distribution that generates autogenous shrinkage instead of self-desiccation in the cement matrix [60].

### 2.7. Effect of Ratio of Activator Liquid to Fly Ash/Slag

If the activator to binder ratio decreases from 0.40 to 0.35, then the fresh mix's workability decreases, but extra water enhances the workability and strength are reduced. The strength reached up to 51 MPa at 28 days of GPC, containing 20% slag and 80% fly ash as a binder and cured at 20 °C [22]. In the fly ash-slag based GPC, the slag/fly ash plays an essential role in the mechanism of the geopolymerisation for strength development. If the slag content in the mix design is over 50% of the binder, this demonstrates the calcium silicate hydrate gel's primary reaction with the Na and Al, leading to the formation of the C–N–A–S–H bond. If increasing the fly ash content in the binder, then the form of N–C–A–S–H type gel as a primary reaction bound the water tightly in the composition and got a higher degree of crosslinking compared to the composition formed of C–A–S–H type gel in the slag binder. This study provides critical evidence of the different mechanisms of the geopolymerisation based on the slag/fly ash ratio content as a binder in the mix design. If the small dosage of the slag in the fly ash is in the mix, the reactions slow down the formation of the C–A–S–H binding gel and the formation of the hybrid C–N–A–S–H gel due to the overtime release of Si and Al in the reactions. Increasing the dosage of fly ash in the mix promotes the zeolite development in the hybrid gels after 28 days of higher temperature curing with a lower activator concentration. After the long curing period, the gel microstructure formation predicted the mechanical capability and durability performance of the GPC samples [61]. If the 15% ACS replaces the GGBFS in GPC, it leads

to the thermal stability of the GGBFS-based GPC up to 1000 °C and shows the amorphous nature to resist the heat treatments used in the formation of refractory bricks. Increasing the dosage of ACS by up to 25% decreases the thermal stability of the geopolymer. If the GGBFS is replaced by silica fume by 10%, it increases the strength at a high level and shows thermal stability up to 500 °C, whereas the ACS dosage shows thermal stability up to 800 °C [58]. If the dosage of slag and sodium silicate is increased, it increases the autogenous shrinkage and chemical shrinkage, and if the water binder ratio decreases, drying shrinkage decreases and autogenous shrinkage increases in the AFS mortar. The GPC's slag content above 70% in the mix reflects the rapid setting and cracks generated due to the autogenous shrinkage and shows the hydration products' denser matrix. The geopolymer matrix is not affected by using superplastizisers in the mix design [62]. If the GGBFS, OPC, and CH in the fly ash increase the concrete's compressive strength, it is applicable when no extra water is added to the alkaline solution [63].

With the increasing GGBF dosage, the GPC's workability and setting time would decrease. When GGBFS dosage was up to 30% of binder, the design mix got 55 MPa of GPC and 63 MPa of geopolymer mortar at 28 days [39]. In the GGBFS/fly ash-based GPC, the formation of C-S-H gel at 27 °C by activating GGBFS shows minimal interaction between the fly ash and GGBFs due to different reactions occurring. The mix design strength was developed by the C-S-H gel formation products [64]. The presence of the CSH gels and geopolymeric gels in the matrix enhances the system's strength with the usual quantity of GGBFS. The geopolymer matrix's CSH gel formation worked as a micro-aggregate and resulted in excellent mechanical strength [36].

In the GPC, the fly ash to activator ratio plays an essential factor regarding strength and fire-resistant matrix and silicate to hydroxide, binder age and curing period show negligible effects on the early strength of the GPC. The optimum point of strength and fire-resistance is found at $Na_2SiO_3/KOH = 2.5$ and FA/activator = 3.0 [65,66]. In the GPC, water in the fresh concrete mix and removed through evaporation causes a crack [67]. The workability of the fresh concrete increases with the increment of water dosage in the mortar mix. In the GPC, when the molarity of sodium hydroxide is above 10 M, it shows an increment of workability with fewer effects on strength by increasing the water content in the mix. The strength of the GPC decreases with the increase of the water to geopolymer solids ratio by mass. This leads to the loose microstructure of the geopolymer paste in the concrete [68].

## 2.8. ITZ (Interfacial Transition Zone)

The reaction mechanism of ITZ is slightly different from the matrix reaction. In the ITZ reaction, a tremendous number of voids initially appear with the water content at ITZ, but the hydration products fill the voids after the hydration process. The difference between the microstructure of the matrix and ITZ is challenging to identify after the reaction. EDAX results describe K/Al and Si/Al content in the ITZ being found at a higher level than in the bulk matrix; well-developed crystalline is not present in the ITZ, and forms a sponge-like amorphous gel [69]. The ITZ properties of sodium silicate activated mortar are good with very low porosity at the interface, and thermal activation provides early strength even when the reaction slows down to give a large age span before setting time. A high concentration of the alkaline solution is required for the strong ITZ bond between the siliceous aggregate and fly ash [70]. According to SEM and Nano-indentation analysis, there is no ITZ bond between the old cement paste and the geopolymer matrix [71]. The superplasticizer used in the mix design improved the microstructure of the ITZ of concrete. The ITZ thickness directly affects the concrete's compressive strength, and it is affected by the superplasticizer dosage in the mix. If the ITZ thickness decreases, then the compressive strength of the mix increases. The superplasticizer's lower dosage in the mix develops the loose and porous ITZ between the aggregate and binder and decreases the mixed concrete's performance by reducing the compressive strength. When the super-plasticiser's dosage is high, the mix develops a dense ITZ and increases the concrete's performance [72]. Soluble silicate in

the mix as an activator liquid plays an essential factor in developing the ITZ between the aggregate and paste in the GPC. If the soluble silicate quantity is meagre, the mix shows the weak compressive strength of the paste mortar and concrete compared to the high dosage of silicate soluble. The chloride present in the mix shows debonding between aggregate and paste by crystallising paste at ITZ. LWA used as aggregate in GPC shows excellent bonding at ITZ due to its porous and rough aggregate surface. The bonding zone deteriorates at a temperature of 800 °C due to the dehydration of microstructural water, and swelling of the unreacted silicate content in the matrix creates microcracks between the paste and LWA [73]. The liquid-to-binder ratio plays a vital role in the ITZ of GPC [74].

*2.9. Effect of Curing Conditions*

The curing temperature plays a vital role in the setting and hardening of the GPC. The GPC samples cured at higher temperatures obtain concrete in 4 h at ambient temperature, but at an ambient temperature below 10 °C, the setting of fresh concrete takes up to 4 days without losing quality. The GPC specimens cured at the higher temperature reached mechanical strength within 1 day, but the hardening of the fresh concrete at ambient temperature increased quality strength after 28 days compared to strength gain in 1 day. Curing time also plays an essential role in gaining strength with temperature [75]. The compressive strength of the GPC increases with the increasing curing temperature from 30 °C to 90 °C [26,30]. If the curing at the ambient temperature is impossible due to the delayed time of fly ash-based GPC, the temperature for curing is favourable for gaining higher strength, and the long curing time enhances the geopolymerisation process. GPC samples cured at higher temperatures for a long time lead to microcavities developing in the microstructure, creating cracks in the sample due to water evaporation from the matrix [76]. The GPC flexural strength is higher than the OPC concrete of the same compressive strength [63]. In the natural pozzolanic-based GPC, the compressive strength increases with increasing time and temperature. The GPC applications are cured in both conditions; atmospheric pressure up to 100 °C and autoclave curing above 100 °C. It enhances the mixed samples' compressive strength by eliminating the micro-cracks in the samples [77]. If the curing time increases from 6 h to 96 h, it increases the compressive strength of the GPC, but beyond 48 h of curing, the compressive strength of the samples is not significant [30]. Pre-curing at an ambient temperature above 95% humidity at room temperature before heat curing is beneficial for strength development [68]. Temperature curing is responsible when the specimens cured for 1 h at elevated temperatures do not reach remarkable strength. The longer curing time is responsible for the strength development at early periods, by accelerating the reaction rates. The curing age enhances the polymerisation process from 4 h to 96 h and increases the compressive strength, but the strength development of the samples is achieved at 24 h of curing, so there is no need to cure beyond 24 h [78].

The geopolymerisation reaction increases with the increase in curing temperature and develops the early strength of the GPC. The GPC samples cured in the fog show the high absorption of moisture. The end products of GPC had a mostly open microstructure when steam cured [58]. In the GPC, the oven-cured specimens reached 90% strength in 3 days and 28 days of compressive strength, but the ambient-cured samples reached up to 82% strength in 28 days. The ambient-cured specimens' ultimate strength was higher than the oven-cured samples because the strength rate beyond 7 days is not significant [79].

The elastic modulus of GPC is directly affected by curing temperature. The MOE of specimens increases with the curing's increasing temperature up to a limit and elated to the water to binder ratio. The evaporation of the water from the matrix during the temperature curing reduces the elastic modulus of the GPC specimens [80]. Figure 3c shows the effect of curing time on the strength of mixed samples.

*2.10. Effect of Calcium Content*

The calcium content in the slag and OPC is used in the GPC to form the CSH gel with the geopolymeric gel at low alkaline conditions and improve the mix's compressive

strength. The lower calcium content available to less CSH gel resulted in the mix's lower overall strength. The calcium content plays a minor role in strength improvement in the high alkaline conditions and forms the CSH gels' precipitation [81]. The high calcium BA's mechanical strength depends on the fineness of the raw sample of BA and the water content in the mix design [82]. The calcium content present in the mix's slag is essential for both early and more prolonged age. If the reaction rate is slow and low strength development confirmed, the low calcium fly ash is used as a binder with the lower concentration of alkali activator used without heat curing. The C-S-H/C-A-S-H precipitation formation initiated the strength development of the GPC fresh concrete, but in the fly ash GPC, the concrete hardens by forming alumina-silicate precipitation. The free calcium content from slag and fly ash dissolution increases the gel's formation and later develops the hardened concrete's strength [83]. The higher calcium in the mix emphasises the C-A-S-H type gel end product with a chain structure. A high amount of calcium content is attributed to high strength GPC [26,31].

### 2.11. Effect of Superplasticizer Addition

A superplasticizer is used to increase concrete strength by lowering the water content of the mix design. The addition of a superplasticizer to the binder mass of up to 2% improves the workability of fresh GPC while having little effect on the strength of hardened GPC [30,84]. The use of superplasticizers is detrimental to the GPC's high-temperature performance [85]. The superplasticizer based on SNF is very effective for GPC [86]. The naphthalene-based superplasticizer is very useful, improving the workability of a slump by about 136%, while the PCE-based superplasticizer increases the workability by around 145%, but it affects the strength of the product blend specimens, reducing the strength by about 29%. In certain cases, SNF-based superplasticizer has no negative impact on concrete strength [87].

The addition of citric acid and sucrose to fly ash-based GPC is an excellent alternative chemical admixture for improving the paste's rheological properties while increasing its weight. Sucrose acts as a retarder in the GPC mixture, while citric acid speeds up the hardening process. The mixture influenced the compressive strength after the sucrose was added, affecting the paste's porosity. Sucrose-added samples resulted in a relationship between compressive strength and porosity, directly attached to compressive strength. Sucrose could be used as a chemical admixture as a retarder in the GPC [88]. The polycarboxylate-based superplasticizer has a greater retarding effect on the fly ash-slag-based GPC without affecting the paste's heat of hydration and has greater workability than the SNF-based superplasticizer. Increasing the content of PCE-based superplasticizer above 2% has little impact on the growing strength before 7 days, but it can have a detrimental effect on the strength of the GPC. The superplasticizer improved the workability and mechanical strength of SCGC and the microstructure of bonding between the paste and aggregate at GPC's ITZ. The microstructure of ITZ differs due to changes in thickness caused by variations in superplasticizer material in the blend, which also influences the compressive strength of GPC. By decreasing ITZ thickness, increased superplasticizer use improves the compressive strength of SCGC and the engineered quality microstructure. Superplasticizer material greater than 2% was inadequate to produce desirable workability with resistance to segregation.

In contrast, 6% and 7% of superplasticizer contents have the necessary workability properties within the EFNARC limits [89], with 7% producing the essential intensity at all ages and increasing the microstructural properties [71]. Because of its intrinsic resilience in alkaline media, the conventional superplasticizer is commonly used as an additive to the binder in OPC concrete to increase strength, but it degrades the hardened specimen strength in GPC blends. The use of a superplasticizer at high temperatures has a detrimental impact on the GPC mixed specimens. At higher temperatures, the SNF-based superplasticizer had no impact on strength. The high molar NaOH content in the GPC mix configuration creates the helpful SNF-based superplasticizer [5].

### 2.12. Effect of Handling Time

The fresh GPC is easily workable for up to 120 min without any strength deterioration [30]. The workability of the fresh GPC increases with increased hand mixing time up to 30 min [28]. If the mixing time is increased, it drastically retards the setting time of the fresh concrete.

### 2.13. Effect of Silicate and Alumina

In the SEM analysis, if the Si/Al ratio $\leq$ 1.40 is present in the matrix, it shows a clustered dense microstructure with large interconnected pores, and if Si/Al $\geq$ is 1.65, then a homogenous microstructure with tiny pores is present. The matrix gel's microstructure increases with the increase of the silicon content, available when the ratio is $1.40 \leq$ Si/Al $\leq 1.65$. The geopolymer microstructure is affected by the absorption of nitrogen and results in the volume expansion of the matrix. The larger gel volume is responsible for higher compressive load and increases the young modulus when the microstructure of the gel is homogenous at the ratio of Si/Al is 1.65. Therefore, the young modulus depends on both compressive strength and homogeneity in the microstructures of the gel. The mixed specimens' ultimate strength is reduced beyond the Si/Al = 1.90 due to the unreacted silica present in the matrix [90]. The thermal shrinkage increases as the mixed content's Si/Al ratio increases due to dehydration, dehydroxylation, sintering, and resilience [59]. The silica and alumina content play a vital role in the reaction of geopolymerisation. Silica content presents the amorphous end products in the reaction, contributing to the higher compressive strength of the mix design through the denser matrix development. The mechanical properties also increase with silica content and achieve a maximum strength of 65 MPa [91].

In the GPC, the $SiO_2/Al_2O_3$ and $SiO_2/Fe_2O_3$ ratios increase with the curing temperature, increasing the mechanical properties of the GPC. They also reduce the water absorption capacity compared to the OPC concrete. The CaO content present in the mix does not affect the geopolymer matrix's reaction [92]. The molar ratio of $SiO_2/Al_2O_3$ increases to 3.4–3.8, which is highly responsible for the high strength gain at a later age [93]. In the fly ash-based GPC, the Si:Al ratio $\geq$ 5 present in the fly ash activated with sodium silicate shows low to moderate strength at ambient curing, but after the heat, the curing shows excellent dimensional stability and high compressive strength.

When the Si:Al ratio is less than 2, it shows high compressive strength but poor dimensional stability and reduces the strength after heating [94]. In the geopolymerisation reaction, increases in alumina and silica content accelerate the range of 3.20–3.70. The mix alumina content increases neither show any zeolitic phase development nor show the mix samples strength development [95]. The alumina content present in the mix is highly responsible for the setting time of the mix. Increasing the Si/Al ratio leads to a longer setting time. In addition, increases in Al content decrease the strength of the concrete [94]. Figure 3d shows the effect of the $SiO_2/Na_2O$ ratio on the autogenous shrinkage of the geopolymer paste.

## 3. Durability and Other Related Aspect Studies

Geopolymers show a great potential in construction industry for durable structures subjected to extreme environment and for sustainable infrastructure development [96,97]. In the durability studies, the long-term strength of the GPC deteriorates with time in aggressive environmental conditions [98]. Acid attack, seawater conditions, sulphate attack, carbonation of concrete, chloride penetration, alkali-aggregate reactions, and free-thaw conditions were included in the durability studies.

### 3.1. Effect of Sulphate Attack

The magnesium sulphate deteriorates the GPC at a very high level in the calcium-rich geopolymer formed in the end products. It breaks the CSH bond and forms the Mg-SH by replacing the calcium present in the structure. The magnesium-formed structure expanded the volume that created the crack formation in the GPC. At the same time, the sodium

sulphate does not deteriorate the GPC at a very high level. The magnesium sulphate reduces the mechanical properties of the GPC-mixed specimens [99]. The high-calcium BA geopolymer mortar shows excellent resistant properties against sodium sulphate [83]. The Fly ash/GGBFS-based GPC shows a 33% deterioration in mechanical strength and a 0.04% expansion after immersion in magnesium sulphate for 360 days, but the OPC concrete deteriorates to 48% mechanical strength and 0.8 expansion in concrete in the same conditions. In the $Na_2SO_4$, exposure to OPC concrete shows a deterioration of strength and expansion of 30% and 0.412%, respectively, but in the GPC, the strength is increased in the same condition [100]. The clay-fly ash-based GPC is less affected by the sulphate attack on the GPC than the OPC concrete because the clay/fly ash-based GPC contains significantly less calcium in the mix [101]. BFA-based GPC is significantly less susceptible to sulphate exposure after 18 months. The OPC concrete reduces up to 20%, but BFA-based GPC deteriorates up to 4% in strength in the same exposure condition as sodium sulphate [102].

### 3.2. Effect of Acid Attack

The acid attack on the concrete decreases the concrete's performance and strength by reducing the specimens' mass loss in the acidic conditions below the 6.5 pH of the concrete. The sulphuric acid immersion in the exposure condition for 28 days shows the weakening of the concrete, and mass loss increased with the GPC matrix weakening. The loss reduction of concrete increases with the increase in acid content. The GPC shows better stability against acidic conditions than OPC concrete due to less calcium content present in the GPC [103]. In the slag-based geopolymer mortar, the addition of nano-silica increases the micro-structure of cement paste (formation of additional calcium silicate hydrate (CSH) gels in the system) and strength properties [104]. If the pozzolanic content of geopolymer mortar is more than 50%, it shows better durable properties than conventional OPC concrete. It is less affected by acid attacks and chloride penetration in concrete. Therefore, alkali-activated slag/fly ash cement are highly useful in acidic or seawater conditions [105].

### 3.3. Effect of Sea Water

The GPC concrete shows better properties against seawater conditions by reducing the concrete's sulphate and chloride penetration [106]. The fly ash-based GPC shows high compressive strength, tensile and flexural strength, low elasticity, water absorption, drying shrinkage, and sorptivity in seawater conditions. The fly ash–based GPC achieved a strength of 55 MPa after 28 days, outperforming the OPC concrete, and was less susceptible to seawater in the same conditions [107].

### 3.4. Effect on Carbonation

The carbonation reaction rate of the GPC depends on the contents of the mixed design present in the concrete. The fly ash/GGBFS based GPC shows weak resistance to the carbonation reaction due to the pozzolanic binder's activation by the mix's sodium silicate. The carbonation reaction increases the concrete's permeability, which is very hazardous for the concrete's durability [108].

### 3.5. Effect of Alkali-Silica Reaction and Leaching

The RCA dosage increases in conventional concrete reduce the concrete's strength and mechanical properties and lead to leaching in the concrete, but in the GPC, the RCA dosage does not affect the strength at a minimal level and reduces the leaching in the concrete [109]. The ASR susceptibility of fly ash-based GPC is lower than that of OPC concrete [110,111]. The non-wood biomass ash-based GPC shows excellent properties against acidic conditions compared to the OPC concrete because the OPC concrete shows 9% mass loss in 28 days under sulphuric acid conditions, while the biomass ash-based GPC shows less than 2% mass loss in the same conditions [112].

### 3.6. Effect of Elevated Temperature

The aggregate size in the mix design of the concrete plays a vital role under high temperatures. If the maximum aggregate size is less than 10 mm in the mix designs, it shows the explosive spalling of the concrete specimens under high temperatures in both types of concrete: GPC and OPC concrete. The spalling of concrete is prevented using a maximum aggregate size of more than equal to 14 mm in the concrete's design mix. The concrete's spalling is explained by the fracture process zone's size (lp), which varies with the aggregate size. The aggregate size is larger than the lp, also long and healthy because of the crack-tip shielding. The GPC is chemically stable under elevated temperatures, whereas the OPC concrete chemically decomposes and dehydrates under the same conditions and decreases in the evaporation water content, decreasing the spalling probability of the concrete [47]. The geopolymer mortars exhibited better performance at elevated temperatures in comparison to control cement mortar mixture [113]. The Si/Al ratio plays a vital role under the elevated temperature; the strength increases with the Si/Al ratio in the exposure of 800 °C of the mixed samples. The heat-cured specimens above 80 °C show higher stability against the elevated temperature. However, the ambient-cured specimens show lower stability in the same conditions, and potassium-based geopolymer shows higher stability than sodium-based geopolymer in high-temperature conditions [66,67]. The GPC has better stability against elevated temperatures than the OPC concrete and is more porous than the OPC concrete analysed by the sorptivity test. It reduces the risk of the spalling of concrete under high-temperature conditions [114].

The FSGC shows similar trends to those of Portland cement concrete in weight loss under elevated temperatures of 600 °C [115]. The GPC shows a higher degree of transient creep and the OPC pastes below the 250 °C temperature. When the temperature ranges between 250 °C and 550 °C, the geopolymer does not show the transient creep increase while the OPC paste shows the higher transient creep, and the geopolymer increases the elastic modulus. OPC concrete shows a minor change in the elastic modulus [116]. The powder slag of ferrosilicon (FSS), an industrial waste, may be blended for manufacturing light weight GPC to conserve its strength properties under elevated temperature [117]. The electric arc furnace steel slag (EAFSS), barite and ilmenite heavy aggregates-based heavyweight geopolymer concrete is very effective in radiation shielding at high temperatures [118]. The dosage of the MPCM shows the stimulating effects on the thermal performance of the PCC and GPC. The number of microcapsules affects the thermal conductivity and latent heat of concrete. Microcapsule dosage increases concrete porosity and has more substantial effects on GPC than PCC [119]. The geopolymer matrix's thermal conductivity is higher than the OPC pastes and shows that the geopolymer paste's specific heat is less than the PC pastes [120]. The pore size distribution plays a vital role in the FA/M-based GPC under elevated temperatures [121].

### 3.7. Effect on the Bond Strength

The GPC specimens explain a similar cracking pattern to the OPC concrete in the pull-out load test, and both fail in a brittle manner due to the splitting of concrete along with the bonding of concrete and bars. The bond strength increases with the concrete strength and concrete cover in both types of concrete. GPC shows higher bond strength than OPC concrete due to the higher splitting strength of GPC for the same compressive strength [122,123]. The GPC beams with lap-spliced reinforcement show similar failure behaviour to the OPC concrete beams. For both types of concrete, the reinforcement is based on Australian Standards and the ACI code. The bond strengths of the beam-ends specimens have lower strength compared to the direct pull-out tests. The bar size plays a vital role in the bond strength; the bond strength increases with the reduction of the bar size [124]. The bond strength of the sand-coated GFRP-reinforced GPC compared to OPC concrete shows higher failure loads than the OPC concrete [125].

The ultimate strength and crack load increases with the increase of the fibre concentration in the mix designs, and it also reduces the cracking rate in the beam [126]. The

sand-coated GFPR bars are a perfect alternative to the internal reinforcement of GPC structures [127]. The elastic behaviour of the GPC under reinforced beams is similar to the under-reinforced OPC concrete beams. GPC specimens show a more brittle flexural strength than OPC concrete specimens [122,128]. The compressive strength of the GPC increases by 5% with the 1% dosage of the 1% steel fibre in the mix [129]. The load capacity of fly ash/GLSS-based geopolymer concrete column increases with the increment of concrete strength, reduction in the load eccentricity, and increase in the design's longitudinal-reinforcement ratio [130]. The column-designed specimens show a similar failure to the design code AS3600 [131].

### 4. Geopolymer Material Applications

The GPC materials have found number of applications in infrastructure development and different fields [132]. The potential applications of geopolymers in sustainable construction are given in [133,134]. Studies have been conducted on the barriers and challenges in the effective application of GPC and reinforced geoploymer composites [135,136]. The geopolymers have been successfully used for soil stabilization in transportation and geotechnical engineering [137,138]. The geopolymers are most suitable for marine engineering construction due to their corrosive resistance and excellent aggressive environment performances [139]. The geopolymers can be used to construct light weight [140] and blast-resistant structures [141]. The geopolymers are used in buildings for floor heating [142] and for energy saving applications, i.e., to reduce HVAC demand of buildings [143]. GEO-based materials can be a sound choice in construction industries in place of OPC for sustainable development due to their thermal energy storage capacity [144], fire retardant capabilities [145], and electrical and self-sensing characteristics [146]. The geopolymer is cost-effective due to its stable performance against elevated temperatures and is used to replace epoxy resins in structural retrofitting with FRP. It is also used as a cost-efficient lining of the trenches to rehabilitate sewage pipelines [147,148]. The geopolymers have proven to be sound materials for structural retrofitting and rehabilitation of heritage buildings [149,150], for building preservation [151,152], and for damaged structural elements [153].

Geopolymers as aggregates in concrete show better strength and stability compared to natural aggregates [154]. The geopolymers meet the desired requirements as a material for 3D printing construction [155,156]. The 3D-printed GPC can be potentially used in a wide range of structural applications in construction industry due to its sustainable processing [157,158]. Table 1 describes the geopolymer applications as per the Si/Al content ratio in the geopolymer mix [159].

**Table 1.** Geopolymer materials applications [159].

| Si/Al Ratio | Applications |
|:---:|:---|
| 1 | ➢ Bricks<br>➢ Ceramics |
| 2 | ➢ Low $CO_2$ cements and concretes<br>➢ Radioactive and toxic waste encapsulation |
| 3 | ➢ Foundry equipment<br>➢ Thermal insulation materials, 200–1000 °C<br>➢ Tooling for titanium processing |
| >3 | ➢ Sealant for industry, 200–600 °C<br>➢ Tooling for aeronautics SPF aluminium |
| 20–35 | ➢ Fire-resistant and heat resistant fibre composites |

## 5. Sustainability

A review of the sustainability of geopolymer concrete is found in [160,161]. In terms of energy used in the production of concrete constituents, sustainability was increased by using alternative materials and reducing energy consumption [162]. In the production of GPC, industrial solid waste is used as a binder, activated by alkaline chemicals for concrete production [154]. Cement, aggregate (fine and coarse), and water are used to produce the PCC. The GPC production uses fly ash, GGBFS, NaOH, sodium silicate, aggregate (fine and coarse), and water. In India, cement production's mostly dry process generates the energy of 4.2 MJ/kg [163]. The fly ash and GGBFS are solid industrial wastes produced by the thermal power plant and steel plant. The fly ash is directly used in the mix, but the GGBFS requires grinding before use in the mix. NaOH production's embodied energy is 20.5 MJ/kg, and sodium silicate production is 5.37 MJ/kg [164]. There is zero embodied energy in fly ash and water. The coarse aggregate and fine aggregates' embodied energy are 0.22 MJ/kg and 0.02 MJ/kg, respectively. The superplasticizer's embodied energy in SNF-based is 12.6 MJ/kg. The total embodied energy of the OPC concrete is 1897.86 MJ/m$^3$, whereas the embodied energy of the GPC is 1749.21 MJ/m$^3$. The GPC's embodied energy is less compared to the OPC concrete. Table 2 describes the total constituents present in both the mix design and their embodied energy.

**Table 2.** GPC and OPC concrete constituents details.

| | Embodied Energy (MJ/kg) | OPC Concrete | | Geopolymer Concrete | |
|---|---|---|---|---|---|
| | | Mix Content (kg/m$^3$) | Embodied Energy Content (MJ/kg) | Mix Content (kg/m$^3$) | Embodied Energy Content (MJ/kg) |
| OPC | 4.2 | 370 | 1554 | 0.0 | 0.0 |
| Fly ash | 0.0 | 0.0 | 0.0 | 303.75 | 0.0 |
| GGBFS | 0.31 | 0.0 | 0.0 | 101.25 | 31.38 |
| NaOH | 20.5 | 0.0 | 0.0 | 40.5 | 830.25 |
| Na$_2$SiO$_3$ | 5.37 | 0.0 | 0.0 | 101.25 | 543.71 |
| Fine Aggregate | 0.02 | 683 | 13.66 | 683 | 13.66 |
| Coarse Aggregate | 0.22 | 1289 | 283.58 | 1269 | 279.18 |
| Water | 0.0 | 148 | 0.0 | 40.5 | 0.0 |
| Superplasticizer | 12.6 | 3.7 | 46.62 | 4.05 | 51.03 |
| | Total | 2493.7 | 1897.86 | 2543.7 | 1749.21 |

The geopolymer materials promote a circular economy process and sustainable development [165,166]. The use of sustainable material, such as geopolymers, their application and economical production techniques in the construction industry create employment and increase energy efficiency [167]. The economic and commercial manufacturing aspects (social), an important component of sustainability of the GPC, has not received much attention. The cost studies of GPC were undertaken by Mathew et al. [168]. The GPC is more cost-effective than OPC. Weil et al. [169] have studied the cost comparison and cost drivers for OPC concrete and GPC. The cement is the main cost driver in cement concrete while the activators are the cost driver in GPC. The cost-effectiveness of GPC manufacturing is demonstrated by Youssef et al. [170]. Van Deventer et al. [171] have discussed the factors affecting the acceptance of GPC technology in industry. They pointed out that the regulations and supply chain issues barrier must be removed to achieve widespread acceptance of GPC. Shamsaei et al. [172] have reviewed the studies devoted to the obstacles related to commercialization of GPC. They pointed out that economic analysis and social/national standards attitude towards the application of GPC requires further research. The 3D-printed geopolymer concrete is cheaper as compared to 3D-printed cement concrete with the same properties [173]. It can be concluded that sustainable development is achieved by employing geopolymers in Indian construction industries because it results in lower CO$_2$ emissions, optimum utilization of natural resources, utilization of waste

materials, economic and long life infrastructure construction, and societal income and employment generation.

## 6. Conclusions

The comprehensive survey of the journals and papers and energy analysis of the OPC concrete and GPC concrete concluded as follows:

- Geopolymers are a perfect alternative to OPC concrete in concrete performance in terms of strength, durable properties, and sustainability.
- GPC reduces carbon footprints by using industrial solid waste like fly ash and slag and reducing cultivated land for dumping.
- GPC has reduced the cost of concrete by the use of industrial waste in the concrete production.
- The GPC shows better mechanical properties compared to OPC concrete.
- The GPC shows better durable properties, for all extreme environmental conditions i.e., acid attack, seawater conditions, sulphate attack, carbonation of concrete, chloride penetration, alkali-aggregate reactions and elevated temperature, than OPC concrete.
- The embodied energy of the GPC is less compared to the OPC concrete for the same compressive strength.
- The GPC materials have found a number of applications in infrastructure development and other various fields, and have become a proven material for the sustainable development in the construction industry.
- The application of GPC and economical production techniques in the construction industry create employment and increase energy efficiency.

## 7. Recommendations and Future Scope

This comprehensive study of the geopolymerisation process shows the development of strength by bonding the raw materials in the reaction process. The comprehensive literature survey shows that the GPC has better strength and durability properties than OPC concrete. It also utilises industrial solid waste and reduces land dumping costs. The following aspects of GPC require further research:

- The requirement to implement GPC on major structural projects such as roads, bridges, buildings, hospitals, and other structures.
- More research on the other non-ferrous solid waste used in the GPC and analysis of such material's capacity to work as a binder in the GPC.
- Further research on the hazards associated with some hazardous material components of GPC for its use on a large scale in the construction sector, and research on ways to reduce the hazard vulnerability in production/application of GPC.
- Further research work on long term effect on strength and durability characteristics.
- Further study on social/national standards attitude towards the application of GPC.
- Further research work to make geopolymer 3D printing process a viable construction approach.
- Further research work to draw relationships between composition, structure, and strength characteristics.

**Author Contributions:** Conceptualization, M.V. and N.D.; methodology, M.N.; validation, I.R. and M.A.; formal analysis, M.V. and M.N.; investigation, M.V.; resources, I.R. and J.M.; data curation, M.V.; writing—original draft preparation, M.V.; writing—review and editing, I.R. and M.A.; supervision, N.D.; project administration, M.A.; funding acquisition, J.M. All authors have read and agreed to the published version of the manuscript.

**Funding:** The authors extend their appreciation to the Deanship of Scientific Research at King Khalid University for funding this work through Research Group (small) Project under grant number [R.G.P1/90/1442].

**Institutional Review Board Statement:** This study did not involve humans or animals.

**Informed Consent Statement:** As a corresponding author or on behalf of all research paper authors, I consent to participate. All author of the research paper is consent to the publication.

**Data Availability Statement:** The available data had been used.

**Acknowledgments:** The authors acknowledge the Dean of the Faculty of Engineering for his valuable support and help.

**Conflicts of Interest:** The authors declare no conflict of interest. The funders had no role in the design of the study; in the collection, analyses, or interpretation of data; in the writing of the manuscript, or in the decision to publish the results.

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
