# Peer review of "Geopolymer Concrete: A Material for Sustainable Development in Indian Construction Industries"

_crystals, doi:10.3390/cryst12040514_

Round 1

Reviewer 1 Report

Dear authors,

Enclosed please find the review report.

Reviewer 2 Report

Alot has been written and review on this subject. Therefore, considerable improvement in presentation and addition in literature is needed to make it a good quality paper. Following are my comments:

Introduction:

1. Number of recent works has been missing in the text. Please add the following references into the text:

A state-of-the-art review on the durability of geopolymer concrete for sustainable structures and infrastructure. Construction and Building Materials. 2021

Durability performance of reinforced waste-based geopolymer foam concrete under exposure to various corrosive environments. Case studies in construction materials. 2022.

A state-of-the-art on development of geopolymer concrete and its field
applications", Case Studies in Construction Materials, 2022

2. Please check the similarity of the text and rephrase the text accordingly.

3. I am quite confused with the heading arrangement- What do you mean by constituent materials? I think authors need to clarify the constituent materials of GPC. 

4. Similarly, like point 3, it goes with the other headings of the paper.

5. Geopolymer application section is too small, think of adding more materials into it. 

6. Heading sustainability, need adding few more references to it.

7. Conclusion needs to be presented in dot points and should contain all the sections in sumarised form. 

Reviewer 3 Report

The manuscript under the title: “Geopolymer Concrete: A Material for Sustainable Development in the Construction Industry” has content that is not fully relevant to the planned title. The aspect of “Sustainable Development “ is not properly described in the manuscript. Moreover, the manuscript mainly based on data for India, and in my opinion it should be also reflected in the title.

The manuscript requires much additional work before publication. In the current form it is not relevant for the Crystals journal and it required to be strongly corrected.

The following elements required to be improved::

  • The title should be connected to the content given in the article.
  • Abstract: The main aim of the article should be clearly formulated.
  • Introduction: The information should be given on a world scale and not only in one country (India) or it should be market in the title and abstract that the article is connected with this area.
  • Introduction: The given numbers should be supported by proper references (especially two first paragraphs.
  • Introduction: lines 52 – 56 and Figure 1. Explain the source of data. Reference no. 3 is related to the publication form 2016. The data given in text and in Figure 1 are from 2019.
  • Figure 1 requires a source of information.
  • Introduction: lack of sources for given formula.
  • Introduction (and all article): please use down index (line 76 and others).
  • Last paragraph of the Introduction part should explain the main aim of the article and stress the novelty aspect / importance given by review.
  • Lack of information on methodology and research methods. In the review article short information about used keywords, databased or general methodology for the literature review should be presented.
  • Parts 2.1. add information about the classes of FA according to ASTM norm.
  • Part 3. Please give a precise title technical aspect of (define)?
  • Wrong numeration of figures. Figure 2 is given 2 times in the text.
  • All text: please give the fill name when you use abbreviation first time, for example, line 209, but also other places in text.
  • References should be given directly after author name – for example line 440.
  • Figure 3b is poor quality.
  • Table 1 requires verification and source of this data. The heat resistance appeared a few times; fibreglass is treated separately from fibre composite. The content is not coherent.
  • Part 5. Requires to be supplemented, please see: DOI: 10.1515/rams-2021-0067 and DOI: 10.3390/ma14164599. What about the application for 3D printing?
  • There is a lack of a chapter on the influence of geopolymers on sustainable development. Part 6 is very generic and requires to be up-dated and supplemented about other research.
  • Conclusions are not always connected with the article content; there should be more in-depth.
  • References: small number of up-to-date literature (from last 5 years).

Round 2

Reviewer 1 Report

The authors responded to all comments. the paper has been improved. I recommend to accept the paper for publication. 

Author Response

Dear Editor,

Authors thank the reviewer for his encouraging comment. Authors are motivated as reviewer find MS up to his/her expectations.

Thanks and regards,

Dr Mohd. Ahmed

Reviewer 2 Report

My comments are addressed. 

However, proof read is needed.

Author Response

Dear Editor,

Authors thank the reviewer for his encouraging comment. Authors are motivated as reviewer find MS up to his/her expectations.

Thanks, Regards,

Dr Mohd. Ahmed

Reviewer 3 Report

The manuscript entitled: “Geopolymer Concrete: A Material for Sustainable Development in the Construction Industry” was improved, however, some points require clarification, including:

  • The title reflects the content given in the article, especially the aspect of “Sustainable Development “ is not properly described in the manuscript. Moreover there is not information about other developing countries than India.
  • Abstract: There is lack of pointed the main aim of the article in the abstract part.
  • Introduction: The information should be given for all developing countreis, or justify why only in country (India) was selected. 
  • 1.2. Definition of abbreviations used in the MS - "MS" is not explained.
  • Parts 2.1. lack of information about the classes of FA according to ASTM norm.
  • Part 3. Please consider change of the sub-title. In this chapter there are not only durability aspects described.
  • All text: the formating is required.
  • Table 1. The heat resistance appeared a few times - please explain.
  • Part 6 Sustability is very generic and requires additional description of influence of geopolymers on sustainable development.
